# A New Design Model of an MR Shock Absorber for Aircraft Landing Gear Systems Considering Major and Minor Pressure Losses: Experimental Validation

Byung-Hyuk Kang [1], Jai-Hyuk Hwang [2] and Seung-Bok Choi [3,*]

1 Department of Mechanical Engineering, Inha University, Incheon 22212, Korea; 1357op@gmail.com
2 School of Aerospace and Mechanical Engineering, Korea Aerospace University, Goyang 10540, Korea; jhhwang@kau.ac.kr
3 Department of Mechanical Engineering, The State University of New York, Korea (SUNY Korea), Incheon 21985, Korea
* Correspondence: seungbok.choi@sunykorea.ac.kr

**Featured Application: Aircraft Shock Absorber with Controllable Damping Force.**

**Abstract:** This work presents a novel design model of a magnetorheological (MR) fluid-based shock absorber (MR shock absorber in short) that can be applied to an aircraft landing gear system. When an external force acts on an MR shock absorber, pressure loss occurs at the flow path while resisting the fluid flow. During the flow motion, two pressure losses occur: the major loss, which is proportional to the flow rate, and the minor loss, which is proportional to the square of the flow rate. In general, when an MR shock absorber is designed for low stroke velocity systems such as an automotive suspension system, the consideration of the major loss only for the design model is well satisfied by experimental results. However, when an MR shock absorber is applied to dynamic systems that require high stroke velocity, such as aircraft landing gear systems, the minor loss effect becomes significant to the pressure drop. In this work, a new design model for an MR shock absorber, considering both the major and minor pressure losses, is proposed. After formulating a mathematical design model, a prototype of an MR shock absorber is manufactured based on the design parameters of a lightweight aircraft landing gear system. After establishing a drop test for the MR shock absorber, the results of the pressure drop versus stroke/stroke velocity are investigated at different impact energies. It is shown from comparative evaluation that the proposed design model agrees with the experiment much better than the model that considers only the major pressure loss.

**Keywords:** magnetorheological (MR) fluid; MR shock absorber; aircraft landing gear; valve path; major and minor pressure losses; impact energy

## 1. Introduction

The landing gear system of an aircraft consists of the main landing gear and the nose gear. The nose gear is mainly located under the nose of the aircraft. The role of the nose gear is steering the aircraft during takeoff, landing, and taxiing and is not used for direct impact energy dissipation during landing. The main landing gear braces the aircraft's fuselage by dissipating most of the impact energy generated by the weight and sink speed of the aircraft during landing. For a stable landing, the shock absorbers of the main landing gear must be designed to dissipate sufficient shock energy. An oleo-pneumatic shock absorber (abbreviated as oleo-strut) is the most popular one used in modern aircraft landing gear systems. A structure called a metering pin is used inside the oleo-strut to vary the orifice area, and hence the damping force, in response to the piston displacement. Because the oleo-strut is the passive damper, its performance is limited. Therefore, the research on active and semi-active shock absorbers for the aircraft landing gear systems is

being undertaken through several different approaches. A magnetorheological (MR) fluid shock absorber is an attractive candidate to achieve optimal landing performance in almost all landing conditions. In general, a valve mechanism for the fluid flow is used for the MR shock absorber [1]. It is well known that MR fluid reacts to the magnetic field as a mixture of carbonyl-iron powder (CIP) and silicone oil. In the presence of the magnetic field, the in-fluid CIP particles are aligned in the direction of the magnetic field to form chain structures. The structure creates yield stress, resisting the flow of the MR fluid, which is a characteristic of the Bingham-plastic flow [2,3]. Thanks to the Bingham-plastic characteristics in response to the magnetic field, it is possible to achieve the desired damping force, resisting the fluid flow by controlling the magnetic field. Therefore, MR fluid is being actively used for several dynamic devices or systems that require a controllable damping force under different operating conditions. These systems include seismic dampers in civil structures, vehicle suspension systems, seat dampers, and lateral dampers for trains. In order to accurately predict the pressure drop (or damping force) of an MR shock absorber, several design models have been proposed so far [4–8].

One of the critical design parameters of MR shock absorbers is to determine the gap (or flow path) in which the flow resistance of the MR fluid is controlled by the magnetic core. In the flow path, two occur: the major pressure loss and the minor pressure loss. The major loss is caused by the fluid's dynamic viscosity generating a hydraulic resistance force in the gap. On the other hand, the minor loss is caused by inertia due to the density of the fluid generated in the inlet and outlet of the gap, which sharply bents the flow path and the developing region. The faster the operating speed of the MR shock absorber, the greater the effect of the minor loss because it is sensitive to the fluid velocity. In previous works on the design of MR shock absorbers, only the major pressure loss has been considered because the application systems are operated in the low operating speed range (or low stroke velocity), which is less than 0.5 m/s. Recently, several works on MR shock absorbers applicable to aircraft landing gear systems have been proposed. In some works [9,10], MR shock absorbers for aircraft landing gear systems were designed, analyzed, and experimentally validated under low-frequency conditions, and showed suitable damping force. An aircraft landing gear system with an MR shock absorber was designed, and the landing performance via the skyhook controller was evaluated in [11]. Robust adaptive control was proposed for the aircraft landing gear system equipped with an MR damper based on the adaptive hybrid control and sliding mode control algorithm to take account of the parameter uncertainties [12]. Moreover, the authors of this work formulated a full-scale aircraft model with 6-DOF dynamics integrated with an MR shock absorber and evaluated the landing performance through the modified skyhook controller and the inverse model of the mechanical energy [13,14]. However, only the major pressure loss was considered in these works.

Consequently, the technical contribution of this work is to propose a novel mathematical design model for an MR shock absorber, which can be applicable to high-speed operating systems, including aircraft landing gear systems, by considering both the major and minor pressures losses. As a first step to achieve this target, a structural configuration of an MR shock absorber for the aircraft landing gear system is presented, and its working principle is explained. The governing equations of motions are then derived, considering the pressure drop and damping force at compression and rebound conditions. Subsequently, a prototype of the MR shock absorber is designed and manufactured on the basis of the governing equations of motions and the design parameters of a commercial lightweight aircraft landing gear system. To validate the proposed design model, a drop test, which can generate a high stroke velocity of 2.5 m/s, is established, and the results on the pressure drop versus the stroke/stroke velocity are experimentally measured at various impact energy conditions representing different stroke velocities. The results achieved from the design models considering the major loss only, the major and minor loss, and the experiment are compared to validate an excellent agreement of the proposed design model with the experiment.

## 2. Mathematical Modeling for Aircraft MR Shock Absorber

### 2.1. Damping Force of MR Shock Absorber

Figure 1 shows the basic configuration of an aircraft landing gear system and a lumped diagram of the MR shock absorber. For MR shock absorbers, not only the pneumatic and hydraulic force of the oleo-strut, but also the controllable MR force, are added. MR force can change the damping force depending on the input current generated by the controller, which is effective for landing performance. There are several different types of MR shock absorber that can be devised for landing gear systems. In this work, a cylindrical tube type, shown in Figure 2, is proposed.

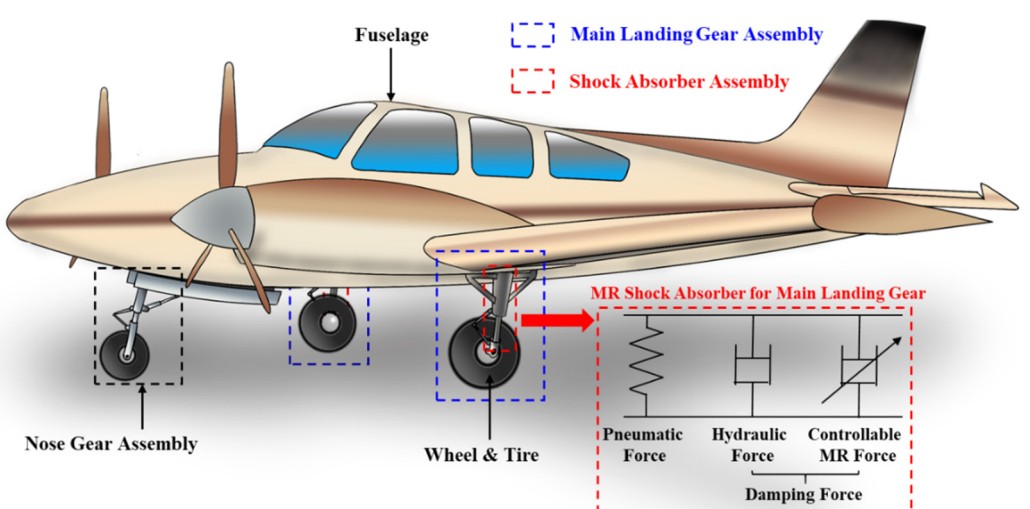

**Figure 1.** Configuration of aircraft landing gear and MR shock absorber.

The left of the diagram is the direction of the aircraft fuselage, and the right is the direction of the ground where the tire and wheel are located. The inside of the shock absorber is divided into an upper chamber, a lower chamber, and a gas chamber, with the MR valve and separator as boundaries. The upper and lower chambers are filled with MR fluid, and the gas chamber can be filled with air or nitrogen. Figure 2b shows a free-body diagram of the internal pressure and the external force acting on the MR shock absorber. From the figure, the main strut of the MR shock absorber is fixed to the aircraft fuselage, and the piston makes a relative motion depending on the stroke displacement, $s$, by the ground reaction. The external force in the direction of the piston axis is called a strut force, $F_S$. When the strut force is applied, the MR fluid flows through the MR valve based on the relative motion of the piston, and the damping force is generated to resist the strut force. In addition, the flow causes a change in the amount of fluid in the lower chamber, and the separator induces a pressure change in the gas chamber while making relative movements inside the piston to compensate for the changing fluid volume and the pneumatic pressure increases. The upper, lower, and gas chamber pressures are defined as $P_1$, $P_2$, and $P_{gas}$, respectively. The MR shock absorber is a form in which a damper and a spring are combined. In other words, the difference between $P_1$ and $P_2$ generates a damping force responsible for the damper's function, and $P_{gas}$ generates the pneumatic pressure, which acts as the gas spring. The difference between $P_1$ and $P_2$ is the pressure drop, $\Delta P$, which can be expressed as follows:

$$\Delta P = P_1 - P_2 \tag{1}$$

where $P_1$ and $P_2$ are the pressure of the upper and lower chambers, respectively. Using Equation (1) and the condition that $P_{gas}$ is equal to $P_2$ in the quasi-equilibrium state,

the strut force, $F_s$, the damping force, $F_d$, and the pneumatic force, $F_{gas}$, are calculated, respectively, as follows:

$$F_s = F_d + F_{gas} \tag{2}$$

$$F_d = \frac{\pi}{4} D_1{}^2 \cdot \Delta P \tag{3}$$

$$F_{gas} = \frac{\pi}{4} D_2{}^2 \cdot P_{gas} \tag{4}$$

where $D_1$ is the inner diameter of the main strut, $D_2$ is the outer diameter of the piston, and $P_{gas}$ is the pneumatic pressure. Due to the volume ratio of the separator and the piston displacements, it is noted that the pneumatic force is expressed as the product of the pneumatic pressure and the cross-sectional area of the outer diameter of the piston, not the inner diameter. In this study, the description of the pneumatic pressure is omitted because the pressure drop for the major and minor losses will be focused on.

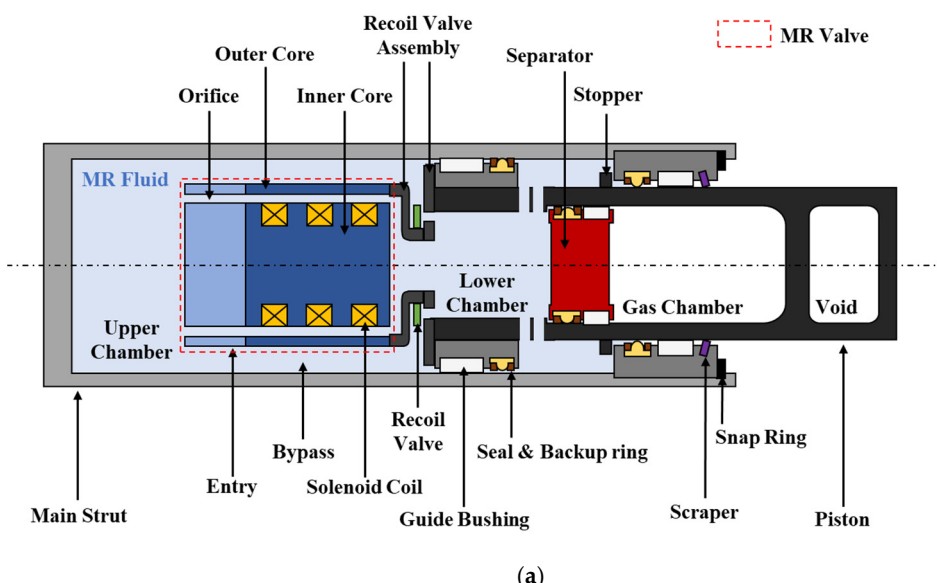

(**a**)

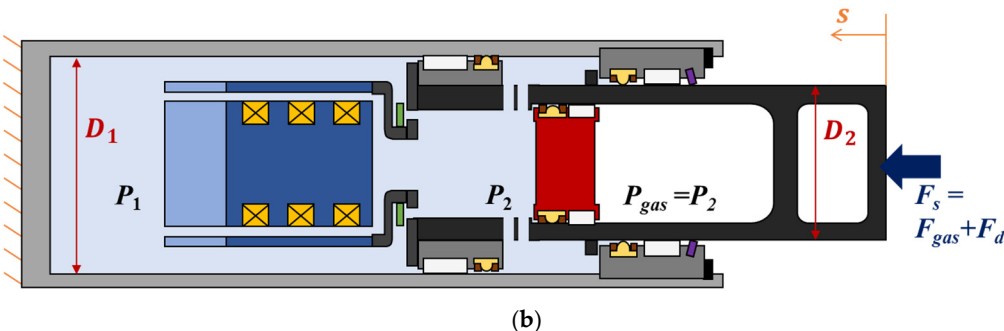

(**b**)

**Figure 2.** Schematic diagram of MR shock absorber: (**a**) configuration; (**b**) free body diagram.

### 2.2. MR Valve Design Considering Pressure Drop of Major and Minor Losses

Figure 3 shows the configuration of the MR valve, the magnetic field direction for the input current, and the pressure drop curves inside the orifice along the *z*-axis. MR valves are divided into entry and MR cores. The MR core is a type of electromagnet that causes the MR fluid to form a magnetic pole by the electric current input of the solenoid coil that is wound around the MR core. The region where the magnetic field path overlaps the MR fluid in the orifice is defined as the magnetic pole. Low carbon steel is mainly used for the MR core to form the path of a magnetic field well. In Figure 3, the state in which the current is applied to the coil is described as on-state, and the state with no current is described as

off-state. The MR core designed in this work has three solenoid coils, and magnetic poles are placed in four locations. In Figure 2b, the pressure of $P_1$ and $P_2$ exist on both sides of the MR valve; the pressure in the region smaller than 0 is $P_1$ and the pressure in the region larger than $L$ is $P_2$, based on the $z$-axis in the graph of Figure 3. The pressure curve for the $z$-axis in the figure represents the total pressure drop under the condition that the piston compresses, where $\Delta P_{major}$ and $\Delta P_{minor}$ are hydraulic pressure drops, considering the major and minor loss, respectively. $\Delta P_y$ indicates the pressure drop generated by the yield stress of the MR fluid. Such a pressure drop produces the flow rate of the MR fluid, and the flow rate, $Q$, is calculated as a function of the piston stroke velocity as follows:

$$Q = \frac{\pi}{4} D_1{}^2 \cdot \dot{s} \tag{5}$$

where $\dot{s}$ is the stroke velocity of the piston. The signs of the stroke velocity and flow rate are set to positive during piston compression and negative during piston rebound. For precise control of controllable force, the fluid must be of a laminar flow inside the MR valve, and the magnetic pole should be located in the fully developed region [4]. When operating at low piston velocity, it is common to design without considering the developing region in Figure 3. However, a general aircraft shock absorber has to dissipate the impact energy generated within 1 s at a sink speed (descent rate) of 3.05 m/s, so its operating speed must be rapid [15]. Therefore, when designing an MR shock absorber for the aircraft landing gear system, the auxiliary part must be installed to locate the core in the fully developed region. The name of this auxiliary part is 'Entry', provided in Figures 2a and 3. The Reynolds number in the annular orifice must be calculated in order to determine the entry length for the annular orifice. The Reynolds number, $Re$, for the annular pipe, taking into account the hydraulic diameter and wetted perimeter, is given by:

$$Re = \frac{2\rho \cdot Q}{\pi \mu \cdot D_o} \tag{6}$$

where $D_o$ is the orifice mean circumference (diameter) and $\rho$ and $\mu$ are density and dynamic viscosity of the MR fluid, respectively; the Reynolds number for the annular pipe must be less than 2000 for laminar flow [16]. In the case of laminar flow, the entry length, $L_e$, can be calculated using the following empirical formula [17]:

$$L_e = 0.0322 \text{max}(Re) \cdot t_o \tag{7}$$

where $t_o$ is orifice gap size. The length of the entry for the MR valve can be determined with the formula. The recoil valve in Figure 2a performs the same function as the check valve and implements asymmetry of the damping force during compression and rebound. Figure 4 shows the flow paths and the orifice and bypass velocity profiles under compressive and rebounding conditions. The recoil valve is closed by flow, only through the orifice during compression (Figure 4a). The recoil valve is opened by parallel flow through the orifice and bypass during rebound (Figure 4b). Figure 4c shows the velocity profile as the fluid flows through the orifice. Because the inner and outer cores do not move relative to each other, the velocity profile is in the form of an annular Poiseuille flow (APF). On the other hand, Figure 4d shows the velocity profile when the fluid flows through the bypass. The relative movement occurs between the inner wall of the main strut and the outer wall of the MR valve in response to the piston stroke, so that the velocity profile is in the form of the annular Poiseuille–Couette flow (APCF). The damping force calculation by APF and APCF is to be determined in Sections 2.3 and 2.4.

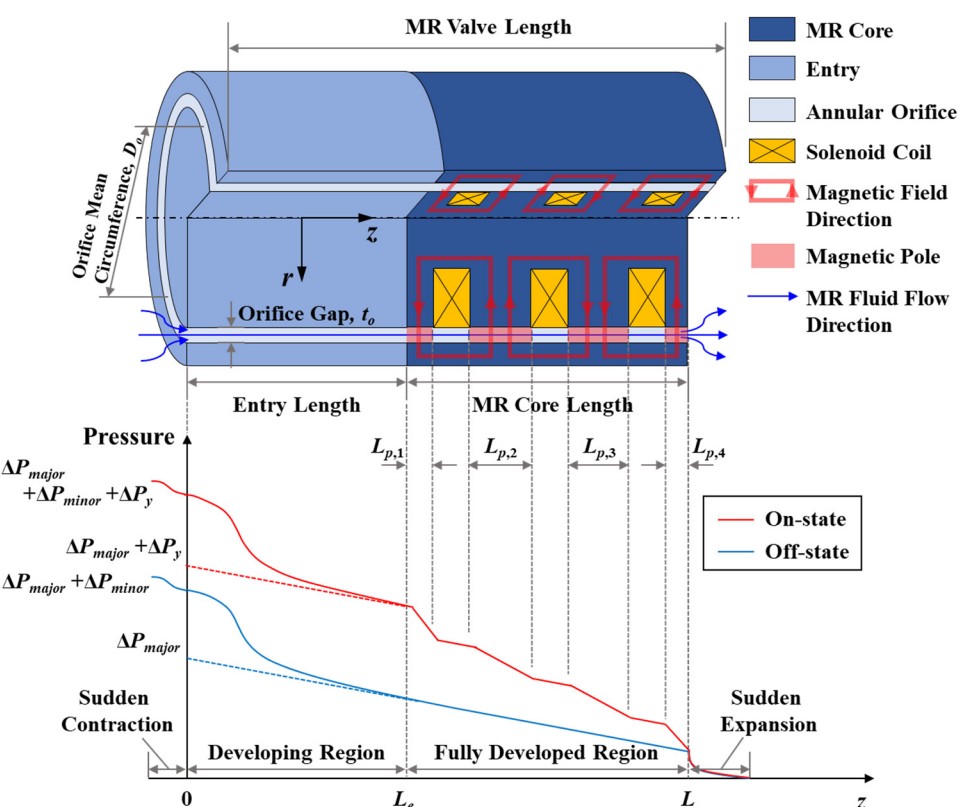

**Figure 3.** MR valve configuration and pressure drop in the annular orifice under compression condition.

### 2.3. Pressure Drop and Damping Force under Compression Conditions

When the piston is compressed, the fluid flows through the flow path, as shown in Figure 4a. Designed to close the recoil valve during piston compression, no fluid flows on the outer wall of the valve, including the bypass. That is, the pressure drop is generated by the velocity profile for off-state shown in Figure 4c because the fluid flows only through the orifice inside the valve. Derived from the Navier–Stokes equation, the pressure drops for the major and minor loss in the case of compression, $\Delta P_{major}$ and $\Delta P_{minor}$, are given by:

$$\Delta P_{major} = C_1 \cdot Q_o \tag{8}$$

$$\Delta P_{minor} = C_2 \cdot Q_o{}^2 \tag{9}$$

where $C_1$ and $C_2$ are the major and minor pressure loss coefficients, respectively, and $Q_o$ represents the flow rate into the orifice. $Q_o$ is equal to $Q$ under compressed conditions because the recoil valve is closed so that the fluid flows only through the orifice. The viscosity causes the major loss, and the inertia of MR fluid causes the minor loss due to the density of the fluid. Therefore, the minor loss cannot be ignored in a system with large fluid inertia such as an aircraft shock absorber, which has a high stroke velocity. $C_1$ and $C_2$ are calculated as follows:

$$C_1 = \frac{1}{8} f \cdot Re \frac{\mu \cdot L}{\pi D_o \cdot t_o{}^3} \tag{10}$$

$$C_2 = \frac{1}{2} \Sigma k_{o,c} \frac{\rho}{(\pi D_o \cdot t_o)^2} \tag{11}$$

where $f$ is the friction factor for laminar flow; $L$ is the orifice and bypass length of the MR valve; and $\Sigma k_{o,c}$ is the total loss coefficient, the sum of loss coefficients determined by the geometric shape of the flow path for orifice flow under compression conditions. The

product of friction factor and Reynolds number, $f \cdot Re$, in Equation (10) can be calculated by the equation of geometry ratio for the annular orifice [18]:

$$f \cdot Re = \frac{128 t_o^2}{D_o^2 + t_o^2 - 2 D_o \cdot t_o / (\ln(D_o + t_o) - \ln(D_o - t_o))} \tag{12}$$

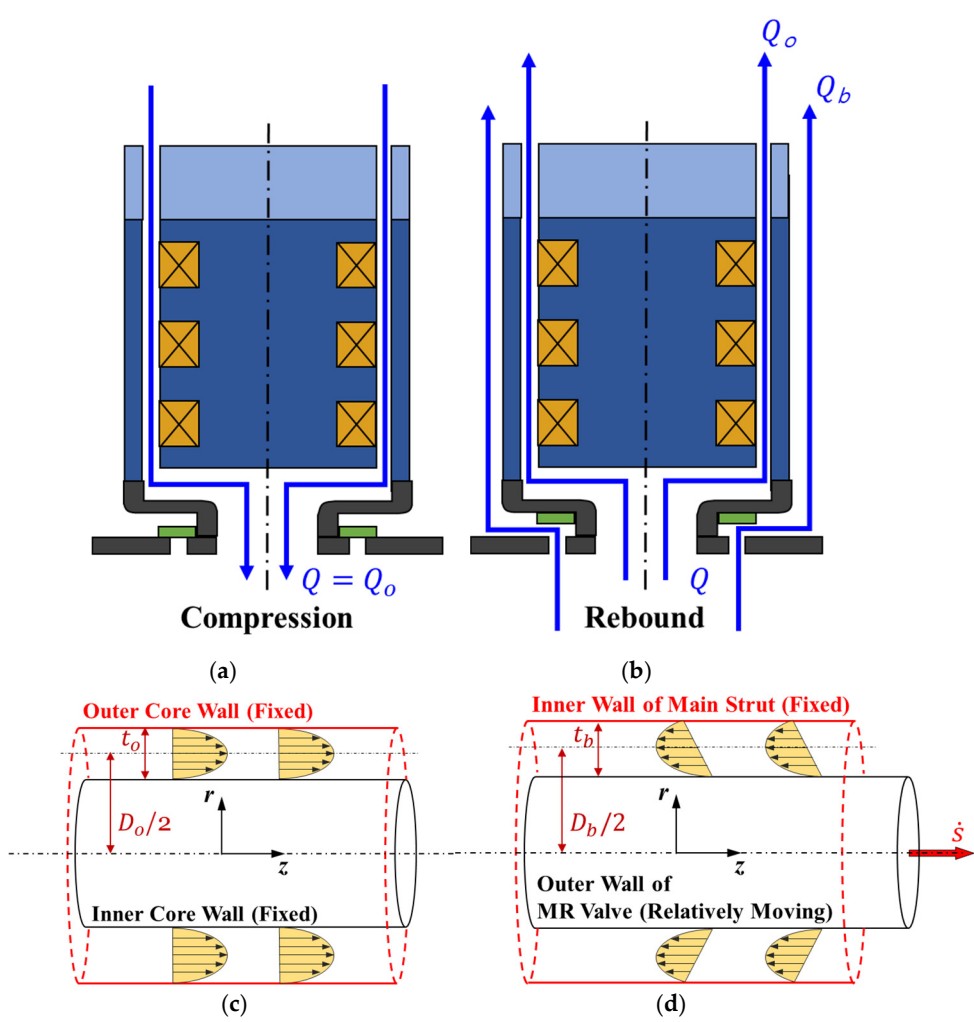

**Figure 4.** Fluid flow in compression and rebound conditions: (**a**) flow path in compression, valve closed; (**b**) flow path in rebound, valve open; (**c**) fluid velocity profile of the orifice: annular Poiseuille flow (APF); (**d**) fluid velocity profile of the bypass: annular Poiseuille–Couette flow (APCF).

As shown in Figure 3, the total loss coefficient from the MR valve is calculated by the sum of the loss coefficient of the sudden contraction, the developing region, and the sudden expansion. MR fluid can be modeled using the Bingham-plastic-viscous model. Therefore, in the off-state case, the pressure drop is originated by the dynamic viscosity and the initial yield stress of MR fluid. In the case of on-state, the magnetic poles generate an additional pressure drop due to the magnetic field, which is defined as the pressure drop due to yield stress. Assuming that the initial yield stress that acts without the magnetic field is infinitesimal, and the core is designed so that all magnetic fields in the magnetic poles act at similar levels depending on the input current, then the pressure drop for the yield stress, $\Delta P_y$, can be expressed by the following simple equation:

$$\Delta P_y = \left( c(\overline{H}) \frac{\tau_y(\overline{H})}{t_o} \sum_j^k L_{p,j} + 3.07 \frac{\tau_0}{t_o} \left( L - \sum_j^k L_{p,j} \right) \right) \text{sgn}(\dot{s}) \tag{13}$$

where $\tau_0$ is the initial yield stress at zero magnetic field, characteristic of the Bingham-plastic-viscous model; $\tau_y(\overline{H})$ is the yield stress, a function of magnetic intensity; $\overline{H}$ represents the mean value of magnetic intensity at all magnetic poles; $k$ is the total number of magnetic poles; and $L_{p,j}$ is the length of the $j$-th magnetic pole; 'sgn' denotes signum function. $c(\overline{H})$ was proposed by G. Yang et al., and the approximated function related to the orifice geometry, the flow rate, and the yield stress are as follows [4]:

$$c(\overline{H}) = 2.07 + \frac{12\mu \cdot |Q_o|}{12\mu \cdot |Q_o| + 0.4\pi D_o \cdot t_o{}^2 \cdot \tau_y(\overline{H})} \tag{14}$$

Because the stroke velocity is positively defined during piston compression, by Equations (8)–(14), the total pressure drop occurring at the orifice in the compression case, $\Delta P_c$, can be summarized as follows:

$$\begin{aligned}\Delta P_c &= \Delta P_{major} + \Delta P_{minor} + \Delta P_y \\ &= C_1 \cdot Q + C_2 \cdot Q^2 + \left( c(\overline{H})\frac{\tau_y(\overline{H})}{t_o} - 3.07\frac{\tau_0}{t_o} \right)\sum_{j}^{k} L_{p,j} + 3.07\frac{\tau_0}{t_o}L \end{aligned} \tag{15}$$

Furthermore, the damping force during piston compression, $F_{d,c}$, can be calculated as follows:

$$F_{d,c} = F_{hyd} + F_{MR} \tag{16}$$

where $F_{hyd}$ and $F_{MR}$ are, respectively, hydraulic force and controllable MR force during piston compression, and they can be expressed as follows:

$$F_{hyd} = \frac{\pi}{4}D_1{}^2 \cdot (\Delta P_{major} + \Delta P_{minor}) \tag{17}$$

$$F_{MR} = \frac{\pi}{4}D_1{}^2 \cdot \Delta P_y \tag{18}$$

### 2.4. Pressure Drop and Damping Force under Rebound Conditions

In the first piston compression during landing, most of the impact energy is dissipated in the shock absorber. Therefore, the hydraulic force during rebound can ignore the minor loss term because the stroke speed is slower than that during piston compression. However, in this work, the effect of minor loss, even in rebound cases, is analyzed. A lower damping force is required for safe landing, which is calculated by comparing the compression case to achieve a high-speed stroke during rebound. The bypass flow path is constructed to implement the lower damping force so that the fluid can flow in parallel, as shown in Figure 4b. When the MR shock absorber rebounds, the parallel flow occurs. Because the recoil valve is opened and fluid flows simultaneously through the orifice and bypass, the flow velocity profiles at the orifice in Figure 4c and bypass in Figure 4d generate the pressure drop. Based on Equations (8)–(14), during the piston rebound, the total pressure drop, $\Delta P_{o,r}$, acting from the orifice by hydraulics and yields is as follows:

$$\Delta P_{o,r} = C_1 \cdot Q_o - C_3 \cdot Q_o{}^2 - 3.07 \left( \frac{\tau_y(\overline{H}) - \tau_0}{t_o}\sum_{j}^{k} L_{p,j} + \frac{\tau_0}{t_o}L \right) \tag{19}$$

where $C_3$ is the minor pressure loss coefficient for orifice flow, expressed similarly to Equation (11), as follows:

$$C_3 = \frac{1}{2}\Sigma k_{o,r}\frac{\rho}{(\pi D_o \cdot t_o)^2} \tag{20}$$

where $\Sigma k_{o,r}$ is the total loss coefficient for the orifice flow under rebound conditions. Because the magnetic field does not act on bypass, the total pressure drop, $\Delta P_{b,r}$ from the bypass can be expressed by considering only the pressure loss of hydraulic and initial yield stress:

$$\Delta P_{b,r} = C_4 \cdot (Q - Q_o) - C_5 \cdot (Q - Q_o)^2 - 3.07 \frac{\tau_0}{t_b} L \tag{21}$$

where $t_b$ is the bypass gap size; $C_4$ and $C_5$ are, respectively, the major and minor pressure loss coefficients for APCF in the bypass. Because MR fluid flows parallel through the orifice and bypass, the coefficients of the major and minor pressure loss of the bypass must be considered. Additionally, APCF at the bypass must be considered because APF and annular Couette flow are formed simultaneously by the relative motion of the stroke from the inner wall of the main strut and MR valve. Figure 4d illustrates the APCF well. The coefficients of major and minor pressure loss for the APCF, $C_4$, and $C_5$, are as follows:

$$C_4 = \frac{12\mu \cdot L}{\pi D_b \cdot t_b{}^3} - \frac{24\mu \cdot L}{(\pi D_1 \cdot t_b)^2} \tag{22}$$

$$C_5 = \frac{1}{2} \Sigma k_{b,r} \frac{\rho}{(\pi D_b \cdot t_b)^2} \tag{23}$$

where $D_b$ is the bypass mean circumference and $\Sigma k_{b,r}$ is the total loss coefficient for bypass flow under rebound conditions. The pressure drop generated in the orifice and bypass during rebound is the same; Equation (19) is equal to Equation (21). Through some mathematical expansion, the flow rate of the orifice, $Q_{o,r}$, and the total pressure drop, $\Delta P_r$ for piston rebound are calculated as:

$$Q_{o,r} = \begin{cases} a_1 + \sqrt{a_1{}^2 + a_2} \mathrm{sgn}(C_5 - C_3) & , \ a_2 < 0 \\ 0 & , \ a_2 \geq 0 \end{cases} \tag{24}$$

$$\Delta P_r = C_4(Q - Q_{o,r}) - C_5(Q - Q_{o,r})^2 - 3.07 \frac{\tau_0}{t_b} L \tag{25}$$

In the above, the variables $a_1$ and $a_2$ are used to simplify the expression and are defined as follows:

$$a_1 := \frac{2C_5 \cdot Q - C_1 - C_4}{2(C_5 - C_3)} \tag{26}$$

$$a_2 := \frac{C_4 \cdot Q - C_5 \cdot Q^2}{C_5 - C_3} + \frac{3.07}{C_5 - C_3} \left( \frac{\tau_y(\overline{H}) - \tau_0}{t_o} \sum_j^k L_{p,j} + \frac{t_b - t_o}{t_b \cdot t_o} \tau_0 \cdot L \right) \tag{27}$$

If the pressure drop caused by the flow rate is smaller than the pressure drop caused by the yield stress, depending on the magnetic field, a block-up phenomenon occurs in the orifice [4]. If this occurs, MR fluid does not flow through the orifice and can only flow through the bypass. Equation (24) well represents the corresponding phenomenon. If only the major loss is considered in calculating the pressure drop in the rebound case, the quadratic term of the flow rate disappears. Based on Equations (19) and (21), the flow rate of the orifice, $Q_{o,maj}$, and the total pressure drop, $\Delta P_{r,maj}$, considering only the major loss for the piston rebound, can be calculated:

$$Q_{o,maj} = \begin{cases} \frac{C_4}{C_1 + C_4} Q + \frac{3.07}{C_1 + C_4} \left( \frac{\tau_y(\overline{H}) - \tau_0}{t_o} \sum_j^k L_{p,j} + \frac{t_b - t_o}{t_b \cdot t_o} \tau_0 \cdot L \right) & , \ Q_{o,maj} < 0 \\ 0 & , \ Q_{o,maj} \geq 0 \end{cases} \tag{28}$$

$$\Delta P_{r,maj} = C_4 \cdot (Q - Q_{o,maj}) - 3.07 \frac{\tau_0}{t_b} L \tag{29}$$

The damping force, $F_{d,r}$, taking into account hydraulics and yield stress in the rebound condition, can be calculated as follows:

$$F_{d,r} = \frac{\pi}{4} D_1{}^2 \cdot \Delta P_r \tag{30}$$

### 2.5. Total Pressure Drop and Damping Force

The total pressure drop, $\Delta P_{total}$, and the damping force, $F_d$, generated in the MR shock absorber, considering both the major and minor losses, are as follows:

$$\Delta P_{total} = \begin{cases} \Delta P_c & Q \geq 0 \\ \Delta P_r & Q < 0 \end{cases} \tag{31}$$

$$F_d = \frac{\pi}{4} D_1{}^2 \cdot \Delta P_{total} = \begin{cases} F_{d,c} & \dot{s} \geq 0 \\ F_{d,r} & \dot{s} < 0 \end{cases} \tag{32}$$

Moreover, the pressure drop, $\Delta P_{maj}$, and damping force, $F_{d,maj}$, generated in the MR shock absorber, considering only the major loss, are as follows:

$$\Delta P_{maj} = \begin{cases} \Delta P_{major} + \Delta P_y & Q \geq 0 \\ \Delta P_{r,maj} & Q < 0 \end{cases} \tag{33}$$

$$F_{d,maj} = \frac{\pi}{4} D_1{}^2 \cdot \Delta P_{maj} \tag{34}$$

## 3. Design Parameters

### 3.1. Characteristic Evaluation of MR Fluid

The damping force of the aircraft shock absorber utilizing MR fluid is sensitive, not only to the density of the fluid, but also to the yield stress for the magnetic field and the dynamic viscosity. Therefore, the dynamic viscosity and the yield stress for the MR fluid are measured through an MR viscometer. The density is calculated by measuring the mass and volume and is 3.510 g/cm$^3$ for the MRF-140CG fluid from LORD Corporation. As a result of measuring via MR viscometer in a temperature environment of 23 degree Celsius, the dynamic viscosity and the initial shear stress of MRF-140CG is estimated to be 0.290 Pa · s and 166.3 Pa, respectively. The yield stress, depending on the magnetic intensity measured by MR viscometer, is fitted to the following polynomial expression:

$$\tau_y(\overline{H}) = -0.005\overline{H}^3 + 0.900\overline{H}^2 + 298.5\overline{H} + 166.3 \tag{35}$$

where the unit of the shear stress and magnetic intensity are Pa and kA/m, respectively.

### 3.2. Magnetic Analysis for MR Core

Magnetic analysis is performed using the ANSYS MAXWELL program. It refers to the magnetic flux density-magnetic intensity curve (BH curve) required for the magnetic analysis of the LORD Corporation's datasheet for MRF-140CG [19]. The corresponding BH curve is used for each material, and the materials of the MR valve and the recoil valve are as follows: inner and outer cores are AISI1008 steel; the entry is aluminum7075-T651; the recoil valve parts are sus304 and aluminum7075-T651; and the coil uses polyester enameled copper wire (PEW) of AWG27 standard. The magnetic analysis conditions are set as follows: the solution type is magnetic transient; the basic mesh number of the orifice is 15; eddy effects and core loss are activated; and nonlinear residual is set as $10^{-6}$. The magnetic intensity for MRF-140CG, obtained by the magnetic analysis, is curve-fitted, and the averaged magnetic intensity is determined as a polynomial equation, as follows:

$$\overline{H} = -6.057I^3 - 0.776I^2 + 57.29I \tag{36}$$

where *I* is the input current in Ampere units, and the unit of magnetic intensity is kA/m.

### 3.3. Minor Pressure Loss Analysis with CFD

It is almost impossible to theoretically calculate the minor pressure coefficients and loss coefficients in Equations (11), (20), and (23). Therefore, to identify the hydraulic resistance, such as the loss coefficient at the design process, direct experiments can be performed, or it is possible to refer to guidebooks experimentally calculated by many researchers for the duct and pipe flow specifications, as they have specific arbitrary cross-sections [20]. However, the loss coefficients for many of the annular MR valves and other parts designed in this study cannot be found in guidebooks. It is also impossible to determine losses experimentally in the design process. Consequently, in this work, the designed model's loss coefficients are estimated using the Fluent program of the ANSYS computational fluid dynamics (CFD) tool. The CFD analysis conditions are set as follows: the solver type is steady-pressure based; the viscous model is a realizable k-epsilon model with model constraints for C2-epsilon, TKE Prandtl number, and TDR Prandtl number set at 1.9, 1, and 1.2, respectively; basic mesh number of the orifice and the bypass are 15; the orifice and the bypass are set as the laminar zone; the solution methods are simple pressure-velocity coupling, least square cell based gradient, standard pressure, second order upwind momentum, first order upwind turbulent kinetic energy, and first order upwind dissipation rate; the residual of absolute criteria is set as $10^{-3}$; the piston stroke speed when the piston is compressed is 2.50 m/s; and the piston stroke speed when the piston is rebounded is 1.33 m/s.

Using Equation (7), the maximum Reynolds number and entry length are calculated at 1113 and 87.8 mm when the sink speed is 3.05 m/s and the input current is not applied. At the same time, it is confirmed that the entry length is about 85 mm as a result of the CFD analysis, as shown in Figure 5, performed under the same conditions. Based on Equations (11) and (23), the pressure drop calculated by CFD includes both major and minor losses, so the total loss coefficients at the orifice and bypass, $\Sigma k_o$ and $\Sigma k_b$, can be estimated as follows:

$$\Sigma k_o = 2\frac{(\pi D_o \cdot t_o)^2}{\rho}\frac{(\Delta P_{CFD} - C_1 \cdot Q_o)}{Q_o{}^2} \tag{37}$$

$$\Sigma k_b = 2\frac{(\pi D_b \cdot t_b)^2}{\rho}\frac{(\Delta P_{CFD} - C_4(Q - Q_o))}{(Q - Q_o)^2} \tag{38}$$

where $\Delta P_{CFD}$ is the CFD result of the pressure drop. Based on the CFD result in Figure 5a, under compression conditions, the entry length and total loss coefficient are set to 85 mm and 2.779, respectively. Through Equations (6) and (7), the Reynolds number and entry lengths of the orifice are calculated under the harshest rebound conditions. The Reynolds number and the entry length of the orifice are 195.31 and 15.41 mm, respectively. The entry length is 11.85% of the total MR valve length. Calculated based on equations in [21], the Reynolds number and the entry length of the bypass are 136.93 and 11.2 mm, respectively. The length of the entry is 8.62% of the total MR valve length. Additionally, under rebound conditions, there is no tendency for additional minor pressure drops in the entry region from Figure 5b. In conclusion, it is assumed that the entry length is short compared to the length of the total MR valve at the time of the rebound. Therefore, there is no need to consider the entry regions for the orifice and the bypass under rebound conditions. The total loss coefficients of the orifice and the bypass are calculated regarding Figure 5b and Equations (37) and (38). The flow rate analysis results of the orifice and the bypass by the CFD analysis to calculate the total loss coefficient are as follows: the flow rates of the orifice and the bypass are 1.886 cm³/s and 1.508 cm³/s, respectively, and the total loss coefficients of the orifice and the bypass are 3.233 and 14.25, respectively.

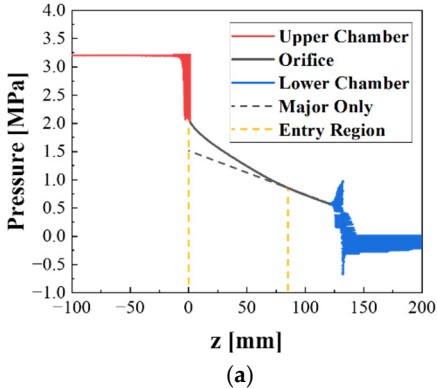 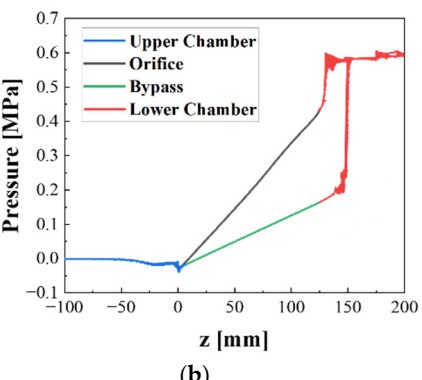

(**a**)　　　　　　　　　　　　　　　　　　　　　　(**b**)

**Figure 5.** Pressure result calculated with CFD tool, ANSYS Fluent: (**a**) pressure change along the *z*-axis, compression condition, flow through the orifice, and comparison with major pressure loss; (**b**) pressure change along the *z*-axis, rebound condition, flow through orifice, and bypass simultaneously.

### 3.4. Design Parameters for MR Shock Absorber

In this work, the design parameters of an MR shock absorber are chosen on the basis of the lightweight landing gear system of a Beechcraft Baron B55 by considering the weight of 230 kg for each main landing gear. It is noted here that the original weight of the Beechcraft Baron B55 is 680 kg for each main landing gear. Hence, the design parameters of the prototype are appropriately adjusted, and the pressure drops are confirmed using the proposed design model. Table 1 provides the detailed design specifications of the MR shock absorber designed and manufactured in this work. It is noted that the optimization is performed on the design variables for the MR core to set the parameters using the equations in [22]. Based on the magnetic field analysis results and design parameters, the controllable forces are calculated as 0.9025 kN, 1.661 kN, and 2.095 at the input current of 0.5 A, 1.0 A, and 1.5 A, respectively, based on a stroke velocity of 2 m/s.

**Table 1.** Specifications of MR Shock absorber.

| Parameter | Symbol | Value | Unit |
|---|---|---|---|
| Orifice diameter | $D_o$ | 43.75 | mm |
| Orifice gap | $t_o$ | 2.450 | mm |
| Bypass diameter | $D_b$ | 54.60 | mm |
| Bypass gap | $t_b$ | 2.400 | mm |
| Length of the orifice and bypass | $L$ | 130.0 | mm |
| Entry length | $L_e$ | 85.00 | mm |
| Length of the pole 1 | $L_{p,1}$ | 5.500 | mm |
| Length of the pole 2 | $L_{p,2}$ | 11.00 | mm |
| Length of the pole 3 | $L_{p,3}$ | 11.00 | mm |
| Length of the pole 4 | $L_{p,4}$ | 5.500 | mm |
| Wire diameter of the solenoid coil | $d_{coil}$ | 0.361 | mm |
| Inner diameter of the solenoid coil | $D_{coil}$ | 20.00 | mm |
| Height of the solenoid coil | $h_{coil}$ | 4.000 | mm |
| Number of turns of the solenoid coil | $N_{turn}$ | 300 | - |
| Inner diameter of the main strut | $D_1$ | 57.00 | mm |
| Outer diameter of the piston | $D_2$ | 50.70 | mm |
| Viscosity of MR fluid | $\mu$ | 0.290 | Pa·s |
| Density of MR fluid | $\rho$ | 3.510 | g/cm$^3$ |
| Initial yield stress of MR fluid | $\tau_0$ | 166.3 | Pa |
| Product of friction factor and Reynolds number for the orifice | $f \cdot Re$ | 95.98 | - |
| Total loss coefficient for the orifice, compression | $\Sigma k_{o,c}$ | 2.779 | - |
| Total loss coefficient for the orifice, rebound | $\Sigma k_{o,r}$ | 3.233 | - |
| Total loss coefficient for the bypass, rebound | $\Sigma k_{b,r}$ | 14.25 | - |

## 4. Experimental Validation

### 4.1. Experimental Apparatus

The configuration of the drop device, the type of sensors, the DAQ device, and the model names used in the experiment are shown in Figure 6. The sprung and un-sprung masses of the constructed drop device are 230 kg and 15 kg, respectively. Experiments are categorized as follows: off-state pressure measurement with impact energy and pressure measurement corresponding to the input current when impact energy is fixed. The impact energy (*IE*) can be calculated through the sink speed at the moment of the touchdown of the un-sprung mass and the distance to the static equilibrium state:

$$IE = (m_s + m_u)(g \cdot z_{1,eq} + 0.5\dot{z}_1) - m_u \cdot g \cdot s_{eq} \tag{39}$$

where $m_s$ and $m_u$ are sprung and un-sprung masses, respectively; $g$ is the gravity acceleration and is set to 9.806; $z_{1,eq}$ and $s_{eq}$ are sprung mass and piston stroke displacements from the moment of touchdown to the static equilibrium state; $\dot{z}_1$ is the sink speed of the landing gear system at the moment of touchdown. The sink speed is estimated through the displacement and the acceleration of the sprung mass, measured with a laser sensor and an acceleration sensor attached to the sprung mass. In other words, the impact energy is converted into the form of the energy dissipated by the pressure loss in the MR shock absorber after impact, the energy dissipated by the friction between the MR shock absorber and the guide shaft of the drop device, and the elastic potential energy stored in the gas chamber and the tire as a spring energy form. This work compared the dissipated energy due to the pressure loss in the MR shock absorber estimated experimentally and the pressure drop model. $\Delta P_{total}$ and $\Delta P_{maj}$ are calculated for the experimentally estimated piston stroke velocities to validate the mathematically modeled pressure drop model. The piston stroke velocity is estimated through stroke displacement, and acceleration is measured via the wire sensor and accelerometers mounted at the sprung and un-sprung masses. $\Delta P_{total}$ and $\Delta P_{maj}$ can be calculated by Equations (31) and (33). In this paper, the pressure drop model, calculated by estimated piston stroke velocity, is called the $\Delta P - \dot{s}$ model. The pressure drop model is constructed in MATLAB, a numerical analysis program. Because the main content of this paper is a comparison of pressure drop in conformity with the presence or absence of the minor loss, the simulation on the landing efficiency using the landing gear model is omitted.

### 4.2. Comparison of Pressure Drop in Conformity with Impact Energy under Off-State

In this section, the drop test is performed by mounting the aircraft landing gear with the MR shock absorber onto the drop device to analyze the internal pressure response in conformity with impact energy under off-state conditions. The impact energy is applied and tested at 456.7 J, 937.5 J, and 1538 J. Figure 7 presents the inside pressure drop and estimated damping force of the MR shock absorber for, respectively, stroke displacement and velocity when the landing gear system is freely dropped. $\Delta P_{total}$ and $\Delta P_{maj}$ on the legend are the pressure drop models calculated using Equations (31) and (33) via the stroke velocity estimated through their respective experiments. Experimental data on the legend refers to the pressure drop measured by the actual measured upper and lower chamber pressures. The initial stroke displacement and velocity start from the origin of the coordinates, as shown in Figure 7. As the compression process progresses, the stroke displacement and velocity, pressure drop, and estimated damping force increase simultaneously. In Figure 7a,c,e, the response of the $\Delta P_{maj}$ model increases in error compared with the $\Delta P_{total}$ model as the impact energy increases. To compare the differences, the results of the dissipated energy for the pressure drop models are presented in Table 2. The difference between the impact energy and the dissipated energy comes from the elastic potential energy conserved through the pneumatic chamber and the tire. In Table 2, as the impact energy increases, the relative error of the dissipated energy is within 15% for the $\Delta P_{total}$ model. However, in the $\Delta P_{maj}$ model, the relative error gradually increases up to 54%. In

Figure 7b,d,f, as the stroke velocity increases, the difference between the $\Delta P_{total}$ model and the experiment does not increase, whereas the error from the experimental result increases compared to the $\Delta P_{maj}$ model; the difference in the slope due to the quadratic term of the flow rate increases as the stroke velocity increases. Table 2 also shows the error results numerically. In addition, Figure 7e,f is suitable for comparing the error of the pressure drop models with a maximum rebound speed of about 0.6 m/s after the first compression; the root mean square (RMS) of errors and relative errors for the experiment and pressure drop models are compared using data with a stroke velocity range of $-0.1$ m/s or less. For the $\Delta P_{total}$ model, the RMS values for the error and relative error with the experimental data are 35.69 kPa and 37.95%, respectively; for the $\Delta P_{maj}$ model, they are 16.22 kPa and 28.92%, respectively. The characteristics of the minor loss are not well displayed during rebound, which resulted in small RMS values of the $\Delta P_{maj}$ model. In other words, the pressure drop model can be simplified as Equations (28) and (29) by considering only the major loss at the time of rebound. In Figure 7e,f, the maximum stroke velocity during the second compression is about 0.5 m/s, which is suitable for comparing the error for the pressure drop model with low-speed compression. The RMS values of the error and relative error for the experiment and pressure drop models are compared using data within the stroke velocity range from 0.1 to 0.5 m/s. For the $\Delta P_{total}$ model, the RMS values for the error and relative error with the experimental data are 10.94 kPa and 4.507%, respectively; for the $\Delta P_{maj}$ model, they are 45.07 kPa and 15.02%, respectively. Furthermore, the dynamic viscosity and initial yield stress are estimated to be 0.2882 Pa·s and 230.3 Pa as a result of performing nonlinear least square parameter estimation based on Equation (15) using the experimental data, where the estimated values are similar to those measured with the MR viscometer. Because the plastic-viscosity model used in the mathematical modeling is constructed based on the measurement data of the MR viscometer, the error level is generally uniform in the low-stroke velocity interval of 0.5 m/s or less. In other words, the design formula for the general MR damper considering only the major loss may be used within an error range of 15% or less in the application utilized in the experiment under the stroke velocity of 0.5 m/s. The maximum pressure drop and the damping force at the maximum compression velocity are compared between the experimental results and the pressure drop models. The RMS of the maximum pressure drop error and relative error are calculated and compared. For the $\Delta P_{total}$ model at the stroke velocity of 0.5 m/s, the RMS values for the error and relative error with the experimental data are 15.49 kPa and 4.104%, respectively, and for the $\Delta P_{maj}$ model they are 54.51 kPa and 14.44%, respectively. Values are 35.13 kPa, 4.997%, 173.1 kPa, and 24.62% in sequence in Figure 7b; 50.79 kPa, 2.412%, 1011 kPa, and 48.01% in sequence in Figure 7d; and 426.6 kPa, 12.41%, 2.032 kPa, and 59.12% in sequence in Figure 7f. Therefore, it is necessary to consider the minor loss as the maximum stroke velocity increases in order to reduce the error.

**Table 2.** Dissipated energy in MR shock absorber for impact energy; RMS values of error for model-based and measured pressure data over stroke velocity range of 0.1 m/s or more.

| Impact Energy (J) | Dissipated Energy | | | | | | Pressure Drop | | | |
| --- | --- | --- | --- | --- | --- | --- | --- | --- | --- | --- |
| | Experiment (J) | $\Delta P - \dot{s}$ Model (J) | | Relative Error (%) | | | RMS Error (kPa) | | RMS Relative Error (%) | |
| | | $\Delta P_{total}$ | $\Delta P_{maj}$ | $\Delta P_{total}$ | $\Delta P_{maj}$ | | $\Delta P_{total}$ | $\Delta P_{maj}$ | $\Delta P_{total}$ | $\Delta P_{maj}$ |
| 456.7 | 276.6 | 263.5 | 200.6 | 4.748 | 27.46 | | 16.72 | 109.2 | 5.650 | 19.37 |
| 937.5 | 737.2 | 672.5 | 406.1 | 8.781 | 44.92 | | 70.28 | 419.1 | 13.54 | 28.07 |
| 1538 | 1339 | 1149 | 621.2 | 14.18 | 53.61 | | 189.5 | 735.8 | 11.66 | 31.72 |

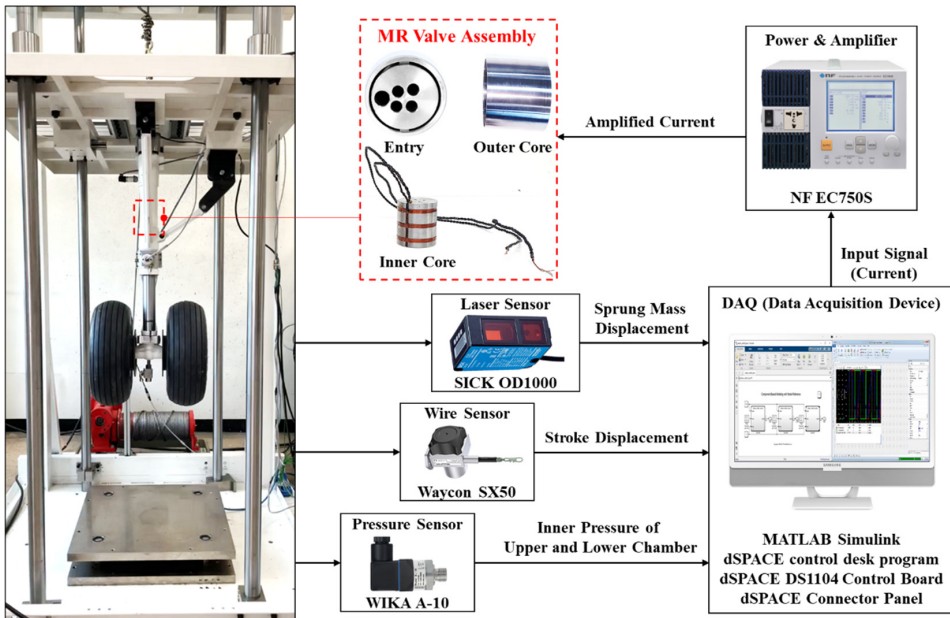

**Figure 6.** Vertical drop test device for performance evaluation of MR shock absorber.

### 4.3. Comparison of Pressure Drop and Landing Efficiency in Conformity with Input Current and Fixed Drop Height

In this section, the response of the MR shock absorber depending on the input current is analyzed experimentally with the impact energy fixed. The input current is applied 10 s before the drop to exclude a time delay with respect to the input current. The impact energy during touchdown is the same at 1538 J. Figure 8 presents the internal pressure drop of the MR shock absorber in conformity with the input current for the stroke displacement and velocity. As in the case of Figure 7, $\Delta P_{total}$ and $\Delta P_{maj}$ on the legend of Figure 8 are the pressure drop calculated for the estimated stroke velocity. 'Experiment' on the legend is the pressure drop measured through the pressure sensors during the drop test. When the impact energy is constant, the pressure drop in all stroke displacement ranges of both the $\Delta P_{total}$ and $\Delta P_{maj}$ models increases as the input current increases. As shown in Figure 8a,c,e, the maximum stroke displacement decreases as the input current increases due to the energy dissipated by the additional pressure drop generated by the yield stress. The dissipated energy is calculated in Table 3 for comparison with the input current. As the input current increases, the dissipated energy for the measured data and the $\Delta P_{total}$ model does not change significantly. Because the impact energy does not change, the potential and kinetic energies do not change, and the dissipated energy also does not. However, the dissipated energy increases as the input current increases in the case of the $\Delta P_{maj}$ model. The reason for this is that the area between the $\Delta P_{total}$ and $\Delta P_{maj}$ curve for the stroke decreases as the dissipated energy for the yield stress increases as the input current increases, as shown in Figure 8a,c,e. Regardless of all the input currents in Figure 8b,d,f, the $\Delta P_{total}$ model does not show a significant difference from the experimental result as the stroke velocity increases. On the other hand, the error between the $\Delta P_{maj}$ model-based pressure drop and the experimental result increase depending on the stroke velocity increase. In Table 3, the RMS values of the error and relative error are calculated over the stroke velocity range of 0.1 m/s or more. The input current and the pressure drop models are independent; the RMS errors and relative errors in Table 3 are due to the error of the magnetic analysis. As a result of the experiment, the controllable force is estimated at 0.6137 kN, 1.382 kN, and 2.128 kN for 0.5 A, 1.0 A, and 1.5 A, respectively, based on the stroke velocity of 2 m/s. The maximum compression velocity estimated by experiments from 0 A to 1.5 A in 0.5 A units is 2.390 m/s, 2.352 m/s, 2.359 m/s, and 2.338 m/s. The error in the estimated velocity occurred due to the friction force of the drop device and MR

shock absorber, but it tends to decrease as the input current increases. The RMS values of the maximum pressure drop error and relative error are compared between the pressure drop model and the experimentally measured data. For the input current, the RMS values of the error and relative error between the experimental data and the $\Delta P_{total}$ model and the RMS values of the error and relative error between the experimental data and the $\Delta P_{maj}$ model are as follows, in sequence, respectively: 0.3399 MPa, 9.484%, 1.889 MPa, and 52.73% for 0.5 A; 0.3052 MPa, 7.922%, 1.863 MPa, and 48.37% for 1.0 A; and 0.3666 MPa, 9.047%, 1.905 MPa, and 47.03% for 1.5 A.

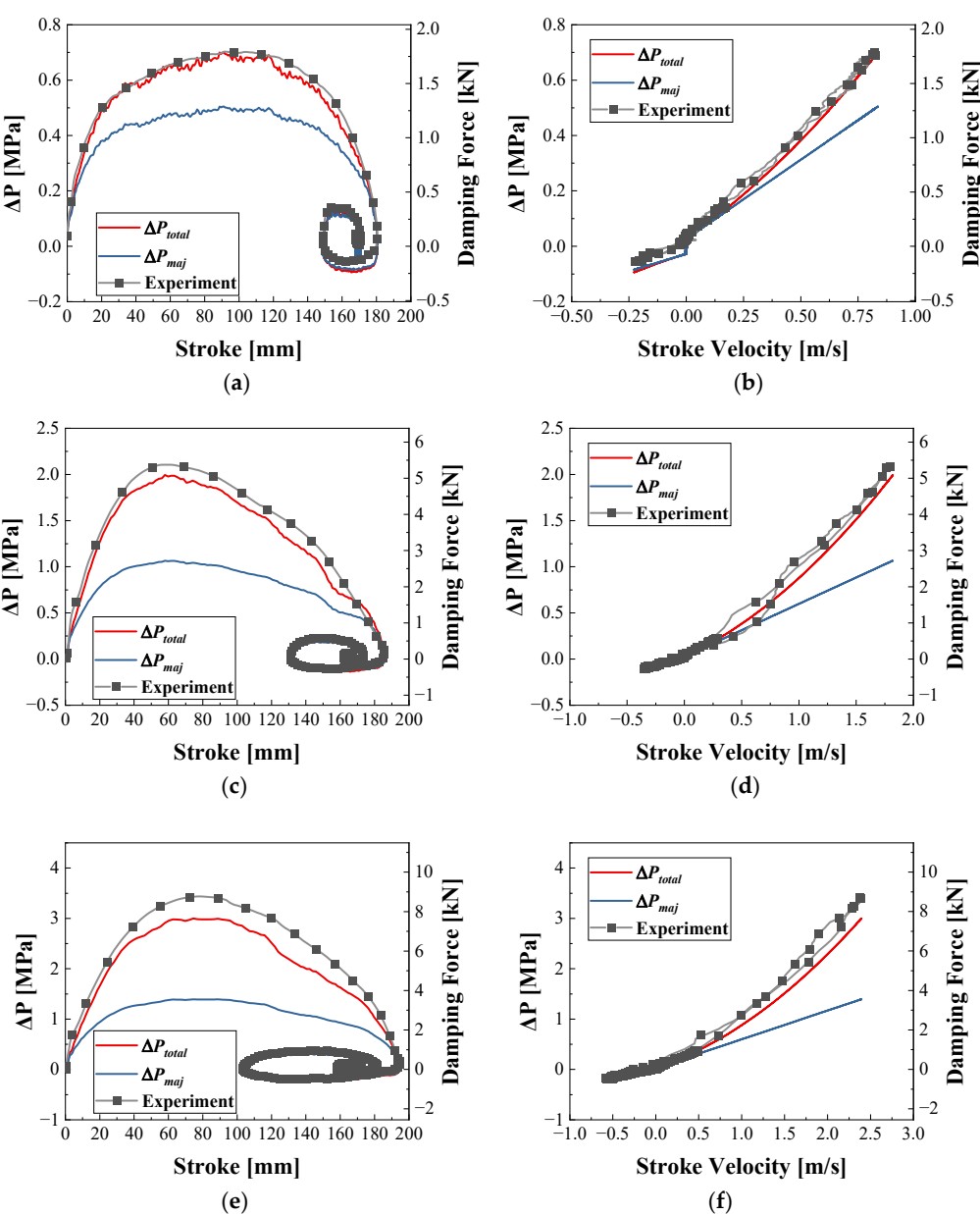

**Figure 7.** Pressure drop and estimated damping force considering $\Delta P_{total}$ and $\Delta P_{maj}$ models and experimental data: (**a**) for stroke and impact energy of 456.7 J; (**b**) for stroke velocity and impact energy of 456.7 J; (**c**) for stroke and impact energy of 937.5 J; (**d**) for stroke velocity and impact energy of 937.5 J; (**e**) for stroke and impact energy of 1538 J; (**f**) for stroke velocity and impact energy of 1538 J.

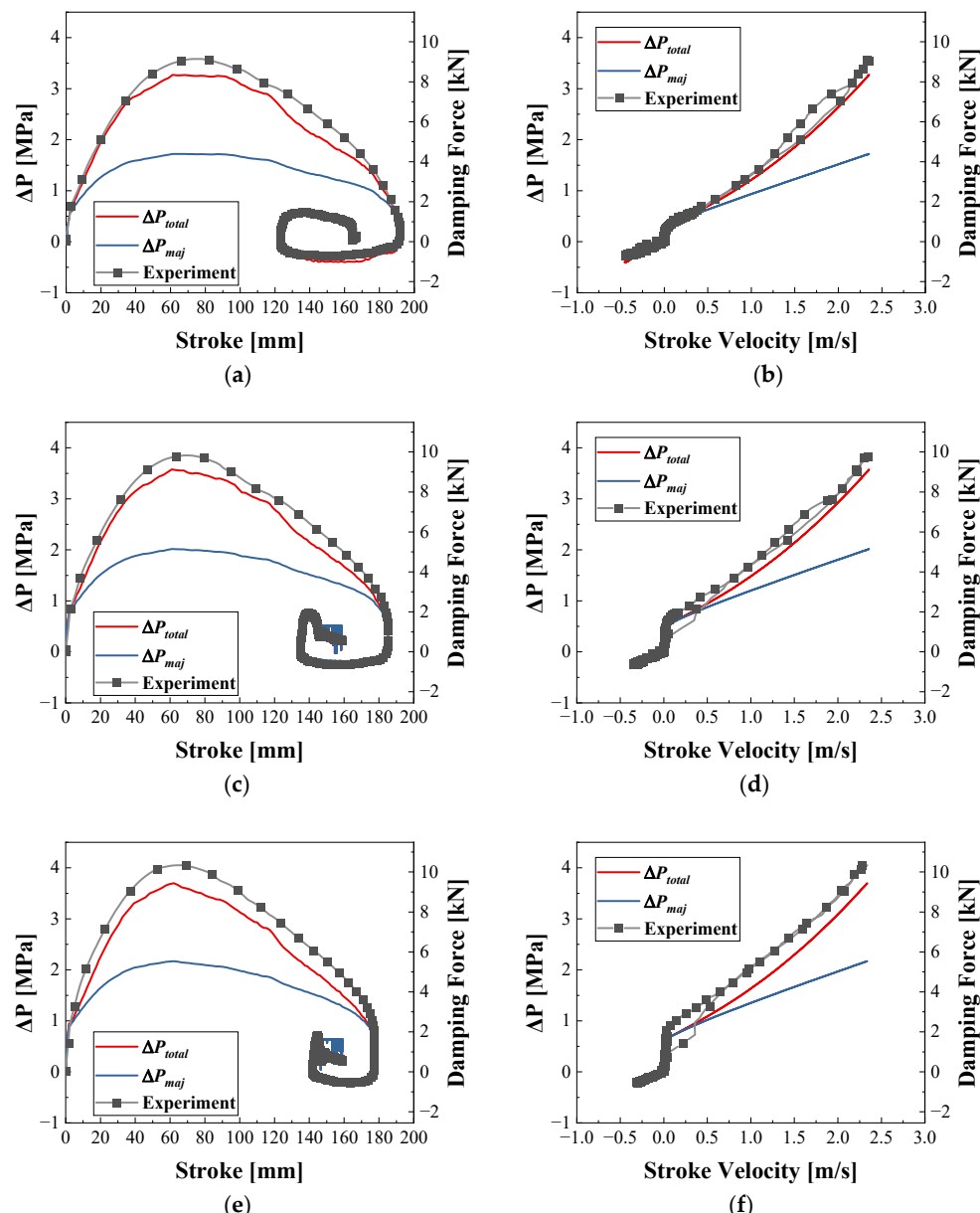

**Figure 8.** Pressure drop and estimated damping force considering $\Delta P_{total}$ and $\Delta P_{maj}$ models and experiment data: (**a**) for stroke and input current of 0.5 A; (**b**) for stroke velocity and input current of 0.5 A; (**c**) for stroke and input current of 1.0 A; (**d**) for stroke velocity and input current of 1.0 A; (**e**) for stroke and input current of 1.5 A; (**f**) for stroke velocity and input current of 1.5 A.

**Table 3.** Dissipated energy in MR shock absorber for input current; RMS values of error for model-based and measured pressure data over a stroke velocity range of 0.1 m/s or more.

| Input Current (A) | Dissipated Energy | | | | | | Pressure Drop | | | |
|---|---|---|---|---|---|---|---|---|---|---|
| | Experiment (J) | $\Delta P - \dot{s}$ Model (J) | | Relative Error (%) | | RMS Error (kPa) | | RMS Relative Error (%) | | |
| | | $\Delta P_{total}$ | $\Delta P_{maj}$ | $\Delta P_{total}$ | $\Delta P_{maj}$ | $\Delta P_{total}$ | $\Delta P_{maj}$ | $\Delta P_{total}$ | $\Delta P_{maj}$ | |
| 0 | 1339 | 1149 | 621.2 | 14.18 | 53.61 | 189.5 | 735.8 | 11.66 | 31.72 | |
| 0.5 | 1377 | 1253 | 757.2 | 8.984 | 45.01 | 172.4 | 749.5 | 10.80 | 28.70 | |
| 1.0 | 1385 | 1253 | 815.7 | 9.564 | 41.10 | 245.6 | 920.5 | 16.36 | 33.66 | |
| 1.5 | 1388 | 1208 | 822.3 | 12.99 | 40.75 | 381.5 | 1087 | 19.08 | 37.02 | |

## 5. Conclusions

In this study, a novel design model for an MR shock absorber for an aircraft landing gear system was proposed, and its effectiveness was experimentally validated. Unlike the conventional model, which features only the major pressure loss, the proposed model was mathematically formulated by considering both the major and minor losses to take account of high-stroke velocity. After deriving the governing equation of motions associated with the pressure drop and damping force, an appropriate size of MR shock absorber was manufactured on the basis of the design parameters of a commercial landing gear system. Because the MR shock absorber for the landing gear is rapidly compressed during a drop, the proposed design model is validated through a drop test, generating different impact energies (stroke velocities). The pressure drop model, with respect to the impact energy and input current, was then compared with the measured result from the drop test. It has been identified from the comparative analysis that the pressure drop model considering of both the major and minor losses show an RMS error of 4.5% compared to the experimentally measured pressure drop. On the other hand, in the case of the pressure drop model considering the major loss only, an RMS error of 15% is identified. This result directly indicates that, when an MR shock absorber is designed for high stroke velocity, both the major and minor losses should be considered to accurately predict the pressure drop and hence the damping force. Finally, in the near future, the proposed model will be used to evaluate the landing efficiency of the aircraft landing gear system with an MR shock absorber in the absence and presence of the feedback controller.

**Author Contributions:** Conceptualization, J.-H.H., S.-B.C. and B.-H.K.; methodology, J.-H.H. and B.-H.K.; data analysis, B.-H.K.; investigation and validation, S.-B.C. and B.-H.K.; writing—original draft preparation, B.-H.K.; writing—review and editing, S.-B.C. All authors have read and agreed to the published version of the manuscript.

**Funding:** This work was supported by the Technology Innovation Program (Intelligent landing gear with variable damping force for 1500 lb class) (10073291) funded by the Ministry of Trade, Industry, and Energy (MOTIE, Korea).

**Institutional Review Board Statement:** Not applicable.

**Informed Consent Statement:** Not applicable.

**Data Availability Statement:** Not applicable.

**Conflicts of Interest:** The authors declare no conflict of interest.

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
