# Peer review of "A New Design Model of an MR Shock Absorber for Aircraft Landing Gear Systems Considering Major and Minor Pressure Losses: Experimental Validation"

_applsci, doi:10.3390/app11177895_

Round 1

Reviewer 1 Report

  1. It is very weird the way the figures 4, 5, 7 and 8 have the letters (a, b,c...) in one side and the images in the other.

Reviewer 2 Report

I believe that the article is scientifically interesting. It can be published but it needs minor additions and clarifications:
1) in the reviewed version of the paper, the figures described as Figures 4b and 4d are missing;
2) the word "laminar" should not be written with a capital letter in the middle of a sentence (see lines 180, 196...). The term "laminar flow" is not named after a person (unlike Reynolds number);
3) is pneumatic fiorce Fgas surely a consequence of pressure Pgas on a surface described by the outer diameter of the piston? 
Why, in this case, is the inner diameter of the chamber containing the Pgas pressure not taken into account?

Reviewer 3 Report

In this paper, a design model of MR shock absorber considering major and minor pressure losses is proposed, and the governing equation of motions are derived at compression and rebound conditions. After manufacturing the prototype, the design model is validated through the drop test generating different impact energies.

The paper contains necessary theory and experimental analysis. I recommend this paper to be published after the following revisions.

  1. In equation (18), ΔPy,c is not explained in the text, please add it.
  2. It can be seen that equation (4) has nothing to do with the Reynolds number and entry lengths of the orifice, please explain ‘Through Eqs. (4) and (5), the Reynolds number and entry lengths of the orifice are calculated under the harshest rebound condition’ in the last sentence of the last paragraph on page 12.
  3. In table 1, the units of viscosity and density of MR fluid are incorrect.
